geology/engineering geology

steep inclination, multi-seam mining, physical model experiment, rock mass failure process, domino destruction

**Author for correspondence:**
Yan Qin
e-mail: qinyancugb@cugb.edu.cn

# Process of overburden failure in steeply inclined multi-seam mining: insights from physical modelling

Hai Wang[1,2,3], Yan Qin[1], Hanbin Wang[1], Yu Chen[1] and Xuancheng Liu[1]

[1]School of Engineering and Technology, China University of Geosciences (Beijing), Xueyuan Road 29, Beijing 100083, People's Republic of China
[2]Shanxi Traffic Layout Reconnaissance Design Institute Co. Ltd, Wuluo Street 27, Taiyuan 030032, People's Republic of China
[3]Engineering and technology research center of highway, Bridge and Tunnel Engineering in Goaf Area of Shanxi, Wuluo Street 27, Taiyuan 030032, People's Republic of China

 YQ, 0000-0001-7344-6378

Ground surface damage caused by steeply inclined coal seam mining is widely distributed in China, but there is little research on the failure process and movement mechanism of strata induced by steeply inclined multi-seam mining. In this paper, a physical model test is carried out to study the failure process and movement mechanism of overburden in steeply inclined multi-seam stepwise mining. The results show that at the initial stage, the main failure of the rock mass is the small-scale collapse at the initial cut and the roof (stability stage of the rock mass). After the roof is exposed over a certain range, the rock mass in the downhill direction slips into the goaf and gradually destroys the interburdens of the goaf, similar to the displacement effect of dominoes (severe failure stage of the rock mass). When the structure of the goaf fails, the overburden subsides, causing extensive damage to the ground surface. The surface damage directly above the goaf is mainly caused by serious subsidence deformation, while the surface damage in the downhill direction is dominated by cracks.

## 1. Introduction

China has large reserves of steeply inclined coal seams, accounting for 15% to 20% of the total coal reserves, and more than 50% of the mines in western China are mining steeply inclined coal seams [1]. The mining of steeply inclined coal seams often results in surface collapse pits and stepped depressions [2], which pose a great

threat to buildings and structures on the ground. Moreover, the movement of the overlying strata and the surface damage caused by underground multi-seam mining are more serious than those caused by single-seam mining [3,4]. Therefore, it is of great significance to study the process of rock mass failure of steeply inclined multi-seam mining for safe mine production and surface subsidence prediction.

According to the difference in coal seam burial form, it can be divided according to the inclination angle [5]: horizontal coal seam (less than 8°), gently inclined coal seam (8–25°), moderately inclined coal seam (25–45°) and steeply inclined coal seam (greater than or equal to 45°).

For the mechanism of overlying strata and surface movement induced by horizontal and gently inclined coal seam mining (less than 25°), predecessors have conducted a great deal of in-depth research on the structural form of overburden, the movement patterns of overburden strata, the control parameters of mining pressure, the breaking patterns of key strata, the change in load over key strata and the characteristics of ground subsidence. By adopting the mechanical model of masonry beam structure, the mechanical model of the elastic foundation beam, the mechanical model of the Kirchhoff plate, the control theory of key strata and some other theories, fruitful results have finally been achieved [6–14].

For moderately inclined coal seam mining, scholars have used numerical simulations, field observations and mechanical models to study the movement characteristics of the overlying strata, the interval of periodic weighting, the patterns of stress distribution, the development of cracks and the prediction of surface subsidence, and have a clear understanding [15–21].

However, research on the process of rock mass failure induced by steeply inclined coal seam mining (greater than or equal to 45°) is not sufficient; in particular, the mechanism of overburden movement induced by multi-seam mining has not been fully revealed. Huayang *et al.* [22] used the vector method to establish a generalized mining subsidence model applicable to coal seams with any dip. Yi *et al.* [23] proposed a prediction model for mining subsidence in inclined coal seams using the flexibility of the influence function. Meng *et al.* [24] used the elastic thin plate theory to establish a mechanical model for roof breaking of the longwall working face in highly inclined coal seams. Deng *et al.* [25] used theoretical analysis and numerical simulation to analyse the stress and deformation process of the roof during the mining of steeply inclined coal seams. Zhang *et al.* [26] used the discrete element method to analyse the deformation and failure processes of the roof along the goaf in a gently inclined thin seam. Zhang & Cao [27] obtained an analytical solution for the deflection and stress distribution of the roof in the steeply inclined working face based on the elastic plate theory. Deng *et al.* [28] used the finite-element method to study the appearance of underground pressure and stress distribution during the process of upward slicing backfill mining in steeply inclined and extra-thick seams. Das *et al.* [29] used the finite difference method to evaluate the influence of the coal seam and strata inclination angle on the stability of the surrounding rock in steeply inclined coal seams. Yun *et al.* [30] used a dynamic monitoring system to monitor the support pressure at the top and bottom of the Dongxia coal mine 37220 working face. Li *et al.* [31] studied the stability of the roof structure of the fully mechanized top coal caving mining face through physical simulation and theoretical analysis. Yin *et al.* [32] used a physical model test to study the movement of coal gangue in the goaf during steeply inclined coal seam mining. Tao *et al.* [33] used a combination of physical model tests and numerical simulations to study the failure mechanism of 45° and 60° inclined layered soft rock strata under deep high gravity and horizontal tectonic stress. Shuyin [34] obtained the formula of the damage depth of the floor under different stress concentration coefficients by establishing a mechanical characteristic model of the plastic failure force of the floor at the working face.

Most of the studies mentioned above are aimed at steeply inclined single-seam mining. While the displacement of surrounding rock is larger in multi-seam mining, and the goaf is very likely to be penetrated, the consequent movement of overlying strata and surface damage is more serious than that in single-seam mining [3,4], but related results are rarely reported. To study the process and mechanism of rock mass failure induced by steeply inclined multi-seam mining, a physical model of steeply inclined multi-seam mining is designed. Through step-by-step excavation to simulate coal mining, photographs and total station monitoring are used to monitor the surrounding rock failure process and movement mode. The process of rock failure during mining is analysed, and the mechanism of surrounding rock movement is discussed. Valuable experience is gained for similar engineering design in the future.

## 2. Establishment of the physical model

The research area of this paper is the Balitang mine of Xinji Colliery, located in the north-central part of Anhui Province, China. The main strata lithologies are sandstone, sandstone-mudstone, sand-mudstone

**Table 1.** Mechanical parameters for rock and physical model materials.

| parameter | unit | sandstone | mudstone | similarity ratio | sandstone in model | mudstone in model |
|---|---|---|---|---|---|---|
| density | kN m$^{-3}$ | 27.4 | 25 | 1.7 | 16.1 | 14.7 |
| compressive strength | MPa | 12.325 | 6.375 | 425 | 0.029 | 0.015 |
| elastic modulus | GPa | 7.225 | 4.675 | 425 | 0.017 | 0.011 |
| cohesion | MPa | 9.35 | 5.525 | 425 | 0.022 | 0.013 |
| internal friction angle | ° | 32 | 25.3 | 1 | 32 | 25.3 |

interlayer and Quaternary deposits. There are four coal seams, numbered #1, #2, #3 and #4 (these numbers are different from the real numbers in the mining area for the convenience of description in this article), and the average thicknesses are 3.55, 2.5, 2.7 and 2.2 m, respectively. The experiment in this paper is carried out under a self-weight stress field, the acceleration similarity constant $C_g$ is 1. Therefore, the main similarity constants include the geometric similarity constant $C_L$ and the density similarity constant $C_\rho$.

$$C_L = \frac{L_p}{L_m} \tag{2.1}$$

and

$$C_\rho = \frac{\rho_p}{\rho_m}. \tag{2.2}$$

$L_p$ is the size of the prototype, $L_m$ is the size of the model, $\rho_p$ is the density of the prototype and $\rho_m$ is the density of the model.

According to the dimensional analysis method, the stress similarity constant $C_\sigma$ should satisfy equation (2.3). Because the unit of elastic modulus and cohesive force is the same as the unit of stress, the elastic modulus similarity constant $C_E$ and the cohesive force similarity constant $C_c$ should satisfy equation (2.4). Meanwhile, the dimensionless parameter similarity constant, including the internal friction angle similarity constant $C_\varphi$, the strain similarity constant $C_\varepsilon$ and the Poisson's ratio similarity constant $C_\mu$, should satisfy equation (2.5).

$$C_\sigma = C_\rho \times C_g \times C_L, \tag{2.3}$$
$$C_E = C_c = C_\sigma \tag{2.4}$$

and
$$C_\varphi = C_\varepsilon = C_\mu = 1. \tag{2.5}$$

According to the rock mechanical parameters data of this mining area and the size of the physical test model frame, the geometric similarity ratio of this test is selected to be 250. The density similarity constant $C_\rho$ is selected to be 1.7, and the stress similarity constant $C_\sigma$ can be calculated using the formula equation (2.3). Sand–plaster–water mixtures were used to simulate the rock strata. The materials similar to the parameters listed in table 1 were obtained by adjusting the material ratio. Uniaxial and shear specimens were made according to the similarity ratio of the model material, and the physical and mechanical parameters were measured. The results showed that the physical and mechanical properties of the material meet the expectations.

According to the geometric similarity ratio and the model frame size, the physical model height is set to 100 cm, the thicknesses of coal seams #1 and #2 are set to 3 cm and the thicknesses of coal seams #3 and #4 are set to 2 cm. The thicknesses of interburden-1 (strata between coal seams #1 and #2), interburden-2 (strata between coal seams #2 and #3) and interburden-3 (strata between coal seams #3 and #4) are 10, 12 and 8 cm, respectively. The inclined formation is gradually laid at a thickness of 1 cm, and mica sheets are evenly spread between each layer of material. Then, hydraulic support is used to rotate the model frame, adjust the rock layer angle to 70° and fill the horizontal sand–mud interbedded rock (thickness 20 cm) and the Quaternary deposits (thickness 20 cm). The detailed dimensions are shown in figure 1. In addition, the thickness in the lateral direction of the model is 10 cm, and the front and back of the model are fixed with acrylic plates.

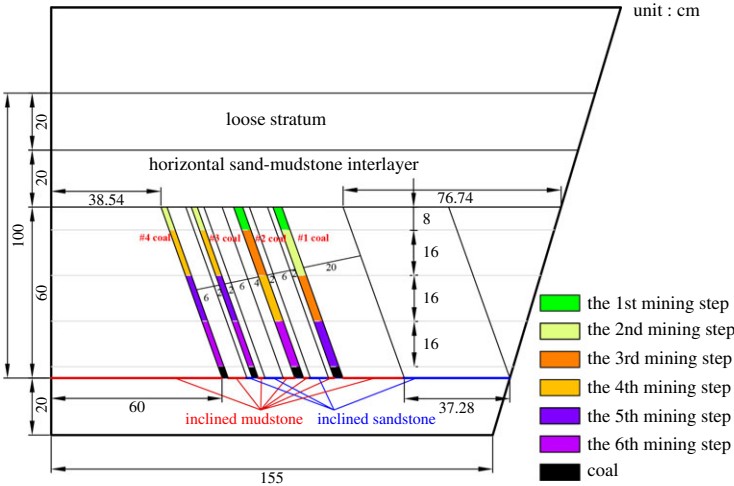

**Figure 1.** Schematic diagram of the physical model strata thickness and mining scheme.

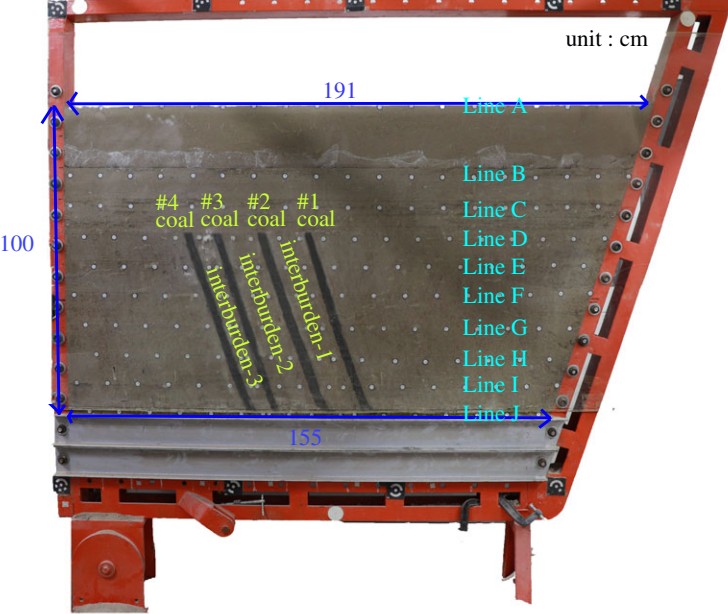

**Figure 2.** Model filling and monitoring point layout.

In this paper, a total station instrument was used to monitor the deformation of the model, and 10 observation lines (Lines A–J) were set up. The distance between measuring points is 10 cm, and the observation line is the same as the dip angle of the inclined rock strata (figure 2). According to the characteristics of steeply inclined coal seam mining, to avoid damaging the adjacent coal seam, the upper and lower seams must be mined simultaneously when the coal seams are close to each other. Therefore, according to the mine stope conditions, the mine in the study area is divided into four co-mining faces. As shown in figure 1, the excavation has six steps according to the designed mining scheme. Only 8 cm is excavated in each excavation step, then excavation stops for at least 1 h to ensure that the rock formation is fully moved, and the total station is used to measure the displacement of the monitoring points.

# 3. Failure process of the overburden

Through the analysis of the physical model test, it is found that the strata movement can be divided into two stages: during the mining process and after mining completion.

## 3.1. Failure characteristics of the overburden during the mining process

### 3.1.1. Mining-induced rock failure process

Multi-seam mining is a step-by-step process, and the description of the phenomenon of each step is complex. To describe the mining area and failure process conveniently, we correlated the coal seam area mined at each step with the test phenomenon and summarized the test results in table 2. The green line in figures (*a*)–(*t*) represents the area of each overburden destruction.

It can be known from the experimental phenomena that during the mining process of coal seam #1, delamination fracture Fr1 was generated in the roof strata first, and as mining progressed, delamination fractures Fr2 and Fr3 continued to be generated. Then, the fractures gradually extended until they penetrated, and the roof strata collapsed. During the mining process of the other coal seams, a small-scale collapse appeared in the initial cut and roof strata, but no delamination fractures occurred. The goafs of coal seams #3 and #4 run through near the initial cut, the model did not cause large-scale collapse and damage, and the rock mass integrity was good. The collapsed broken rock mass moved toward the working face and eventually accumulated near it. Due to the bending effect of interburdens, the horizontal strata above the mined-out area were bent, and a series of cracks were caused on the ground surface.

### 3.1.2. Start-up vertical height of mining collapse

When a steeply inclined coal seam is mined for a certain distance, the immediate roof collapses for the first time, and the vertical height between the working face and the initial cut is defined as the start-up vertical height of mining collapse.

As shown in figure 3, the start-up vertical height of roof collapse is also different for different coal seams. The start-up vertical height of mining collapse of coal seam #1 is 7.5 cm and those of coal seams #2 and #3 are 12.5 and 17.5 cm, respectively. The later the coal seam is mined, the higher the start-up vertical height of mining collapse. However, for coal seam #4, the start-up vertical height of mining collapse is only 10 cm. It is presumed that the immediate roof rock mass of the #4 coal seam has poor properties, which caused the roof to be damaged under its own weight after mining for a certain distance. It can be seen that in steeply inclined multi-seam mining, the more backward the mining is, the larger the initial caving step distance of the coal seam roof is. But in practice, the mechanical properties of rock mass should also be paid more attention to prevent obvious mine pressure disasters.

## 3.2. Failure characteristics of overburden after mining completion

### 3.2.1. Model changes after mining completion

Five hours after the completion of mining (figure 4*a*), the subsidence centre of bending deformation was located directly above coal seam #1. Eleven hours after the completion of mining (figure 4*b*), the horizontal strata above the goaf all collapsed downward and were randomly distributed in the goaf. However, the rock mass in the normal direction of the inclined coal seam moved toward the goaf neatly and orderly, causing the goaf to collapse. The subsidence centre was transferred from directly above coal seam #1 to above coal seam #3. Twenty-one hours after the completion of mining (figure 4*c*), the subsidence of the goaf was still further increasing. As shown in figure 4*c*, the ground surface showed asymmetric subsidence, and the maximum subsidence point was located near the centre of the goaf. The surface influence range was larger in the uphill direction and smaller in the downhill direction. Rock strata failure is mainly tensile failure or shear slip failure.

Three and 5 h after the completion of mining, it was observed that the immediate roof strata of each coal seam still collapsed to a small extent. Eleven hours after the completion of mining (figure 5*a*), a large-scale collapse occurred in the immediate roof strata of coal seam #1 and the horizontal strata above it, and the collapsed rock crushed interburden-1 and then interburden-2 and interburden-3, leading to the large-scale failure of the rock mass in the goaf. The upper horizontal strata and the loose strata sank with the collapse of the lower part, and the ground surface appeared to have a collapse pit. The number of surface cracks increased from five to six, and the lengths increased. Twenty-one hours after the completion of mining (figure 5*b*), the surface subsidence area was further expanded, and a large crack (crack 10) appeared at the ground surface, which was 51 cm away from the right end of the model, with

**Table 2.** Summary of overburden destruction and movement in steeply inclined multi-seam mining. (Note: 'Cr' denotes 'Crack' and 'Fr' denotes 'Fracture'.)

| mining step | figure of current mining area | main phenomenon |
| --- | --- | --- |
| 1st | <br>(a) The red area was mined | After the red area was mined (figure a), the rock mass at the initial cut of coal seam #1 was partially collapsed and nearly arched in shape, with a height of 2 cm.<br><br>After the green area was mined, no deformation was observed. |
| | <br>(b) The red area was mined | After the red area was mined (figure b), the rock mass at the initial cut of coal seam #1 collapsed again by 1.5 cm, the total height reached 3.5 cm, and the width was approximately 3 cm.<br><br>After the green area was mined, no deformation was observed. |

(Continued.)

**Table 2.** (*Continued.*)

| mining step | figure of current mining area | main phenomenon |
| --- | --- | --- |
| 2nd |  (c) The red area was mined | After the red area was mined (figure *c*), the rock mass collapsed at the initial cut, the shape was arched, 1 cm high and 2 cm wide. At this point, two cracks appeared on the surface, as figure *d* shows, and the cracks were located above coal seam #1. Cr1 was 18 cm long and the angle was 68°; Cr2 was 10 cm long and extended down to two directions, and the angles with the vertical direction were approximately 70° and 50°. After the green area was mined, no deformation was observed. |
| 3rd |  (e) The red area was mined | After the red area was mined (figure *e*), the fracture Fr1 appeared in the roof of coal seam #1, with an extension length of 28 cm, and a vertical height of approximately 2.5 cm from the coal seam #1 goaf, the two ends of the fracture tended to extend toward the goaf, and the rock mass at both ends also partially collapsed. After the green area was mined, no deformation was observed. |

(*d*) The red area was mined

(*Continued.*)

**8**

**Table 2.** (*Continued.*)

| mining step | figure of current mining area | main phenomenon |
| --- | --- | --- |

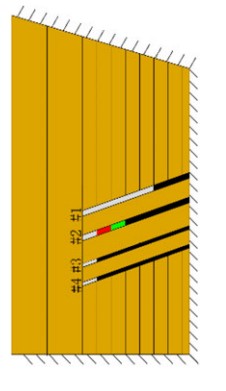

After the red area was mined, no deformation was observed.

After the green area was mined (figure *f*), the second fracture Fr2 appeared in the roof of coal seam #1, which was basically parallel to Fr1, the length of Fr2 was 20 cm. At this point, the width of Fr1 is slightly wider, and the vertical distance between these two fractures is 2 cm.

Approximately 20 min later (figure *g*), it was found that the roof of coal seam #2 collapsed, with a height of 1.5 cm and a length of approximately 10 cm. At the same time, Fr1 and Fr2 were further extended to 31 and 23 cm, respectively.

Approximately 1 h later (figure *h*), the roof of coal seam #2 collapsed again, 2 cm high and 24 cm long. At this point, as figure *i* shows, the roof of coal seam #1 slumped along Fr1, the slumped rock masses were irregularly stacked and filled the goaf, and the maximum collapse was 5.2 cm high and 42 cm long. In addition, the collapse height of the rock strata above coal seam #3 increased from 1 to 2 cm, and the shape was still arched. The cracks Cr1 and Cr2 extended slightly; Cr1 extended to 19 cm, and Cr2 extended to 12 cm. Moreover, the ground surface there had a partial collapse phenomenon. In addition, a new crack Cr3 was generated in the middle of the two cracks, 110 cm away from Cr1 and 16 cm in length. New cracks Cr4 and Cr5 were also generated at the two ends of the model surface. The length of Cr4 and Cr5 was 5 and 8 cm, respectively. Both cracks were almost perpendicular to the ground surface.

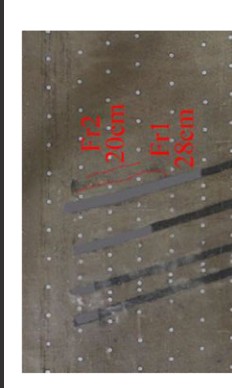

(*f*) The green area was mined

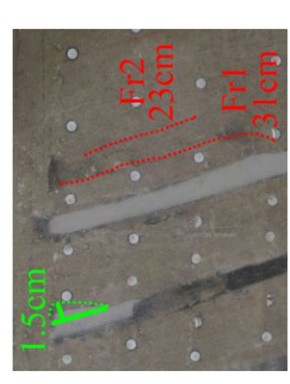

(*g*) 20 min after the green area was mined

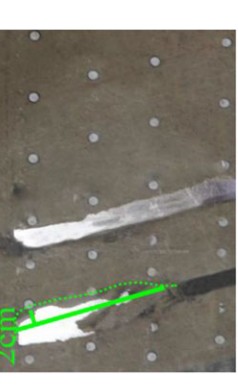

(*h*) 1 h after the green area was mined

**Table 2.** (*Continued.*)

| mining step | figure of current mining area | main phenomenon |
|---|---|---|
| 4th | <br>(*i*) 1 h after the green area was mined | After the red and green areas were mined, no deformation was observed. |
| | <br>(*j*) The green area was mined | After the red area was mined, no deformation was observed.<br><br>After the green area was mined, as figure *j* shows, the three cracks Cr1, Cr2 and Cr3 in the middle of the surface collapsed, and the cracks all extended. The length of the cracks from left to right reached 20, 16.5 and 13 cm, respectively. |

(*Continued.*)

**Table 2.** (*Continued.*)

| mining step | figure of current mining area | main phenomenon |
|---|---|---|
| |  | After the red area was mined (figure *k*), part of the rock mass in the roof strata of coal seam #4 collapsed, approximately 1.2 cm in height and 12 cm in length. After the green area was mined (figure *l*), the roof strata of coal seam #4 continued to collapse, and the collapsed arch further expanded. The collapsed arch was 2.3 cm high and approximately 26 cm long. |

(*k*) The red area was mined

(*l*) The green area was mined

(*Continued.*)

**Table 2.** (*Continued.*)

| mining step | figure of current mining area | main phenomenon |
| --- | --- | --- |
| 5th |  | After the red area was mined, no deformation was observed. After the green area was mined (figure *m*), the roof of the working face collapsed. At the same time, Fr2 in the roof of coal seam #1 showed signs of expansion, and a new crack Fr3 with a length of 12 cm was developed below it. After waiting for 2 h (figure *n*), the roof of coal seam #1 collapsed along Fr2 and Fr3 and was similar to a trapezoidal arch, with a maximum collapse height of 10 cm and a length of 23 cm. At this time, the fractured rock mass above the coal seam #1 goaf all collapsed. |

(*m*) The green area was mined

(*n*) 2 h after the green area was mined

**Table 2.** (*Continued.*)

| mining step | figure of current mining area | main phenomenon |
|---|---|---|
| 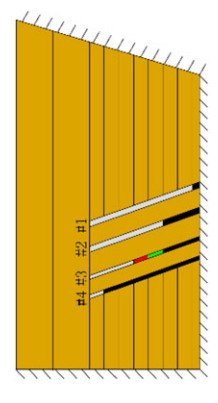 | | After the red and green areas were mined, no deformation was observed. |
| 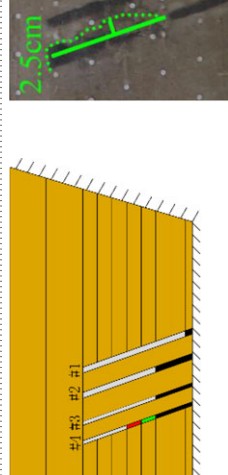 | 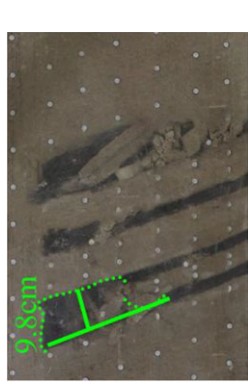<br>(*o*) The red area was mined<br><br>(*p*) The green area was mined | After the red area was mined (figure *o*), the roof of coal seam #4 continued to collapse.<br>After the green area was mined (figure *p*), the upper part of the roof of coal seam #4 collapsed as a whole; due to the close distance between coal seams #3 and #4, the two mined-out areas were connected. |

**13**

**Table 2.** (*Continued.*)

| mining step | figure of current mining area | main phenomenon |
|---|---|---|
| 6th | 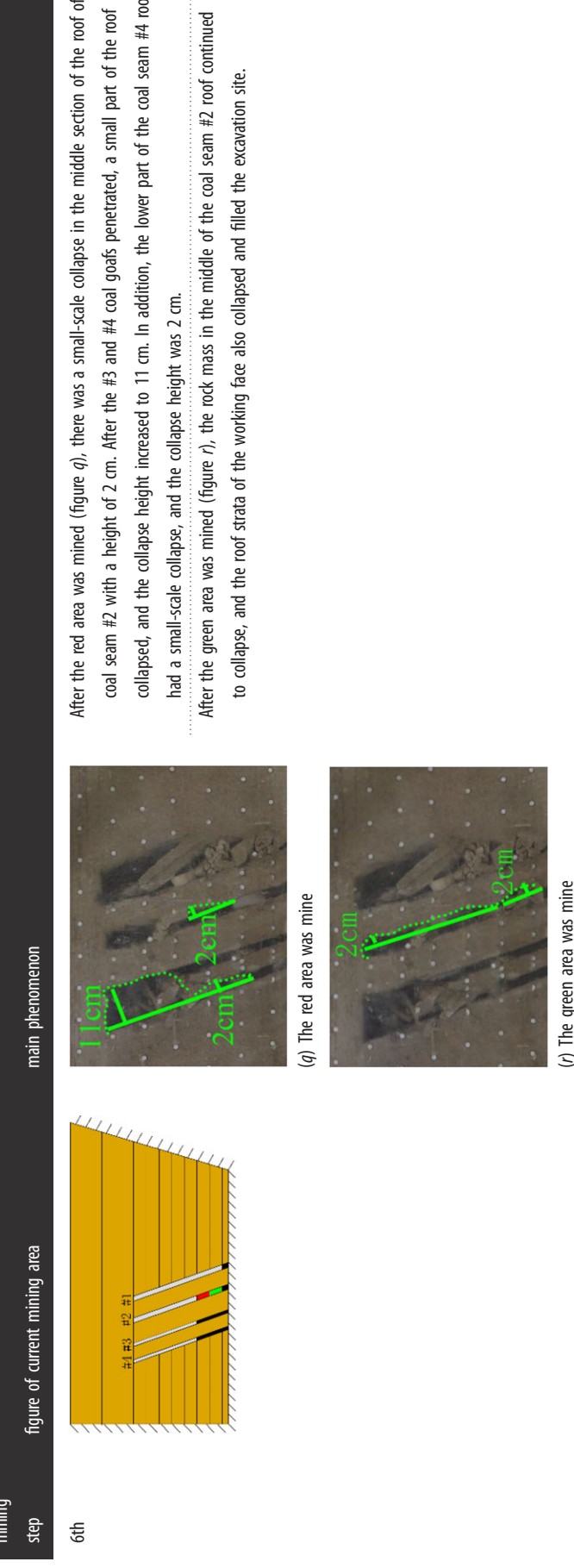 | After the red area was mined (figure *q*), there was a small-scale collapse in the middle section of the roof of coal seam #2 with a height of 2 cm. After the #3 and #4 coal goafs penetrated, a small part of the roof collapsed, and the collapse height increased to 11 cm. In addition, the lower part of the coal seam #4 roof had a small-scale collapse, and the collapse height was 2 cm. |

(*q*) The red area was mine

(*r*) The green area was mine

After the green area was mined (figure *r*), the rock mass in the middle of the coal seam #2 roof continued to collapse, and the roof strata of the working face also collapsed and filled the excavation site.

(*Continued.*)

**Table 2.** (*Continued.*)

| mining step | figure of current mining area | main phenomenon |
| --- | --- | --- |
| | 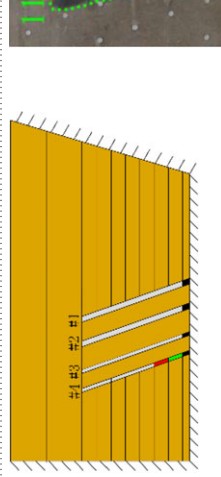 | After the red and green areas were mined, no deformation was observed. |
| | 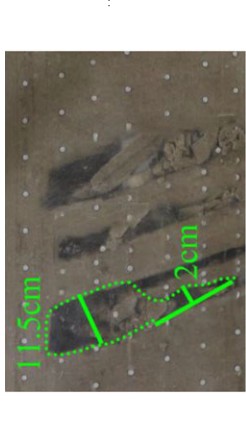<br>(s) The red area was mine<br><br>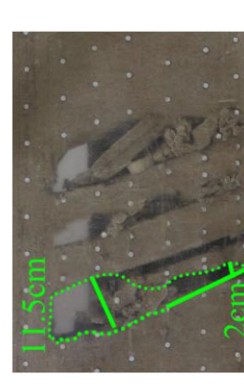<br>(t) The green area was mine | After the red area was mined (figure s), due to the connection of two goafs, the horizontal strata above coal seams #3 and #4 had collapsed as a whole, with a height of 2 cm. The roof strata of the coal seam #4 working face collapsed in the shape of an arch, the collapse height was 2 cm and the length was 10 cm.<br>After the green area was mined (figure t), the roof of the excavation site partially collapsed, falling 2 cm in height and 5 cm in length. |

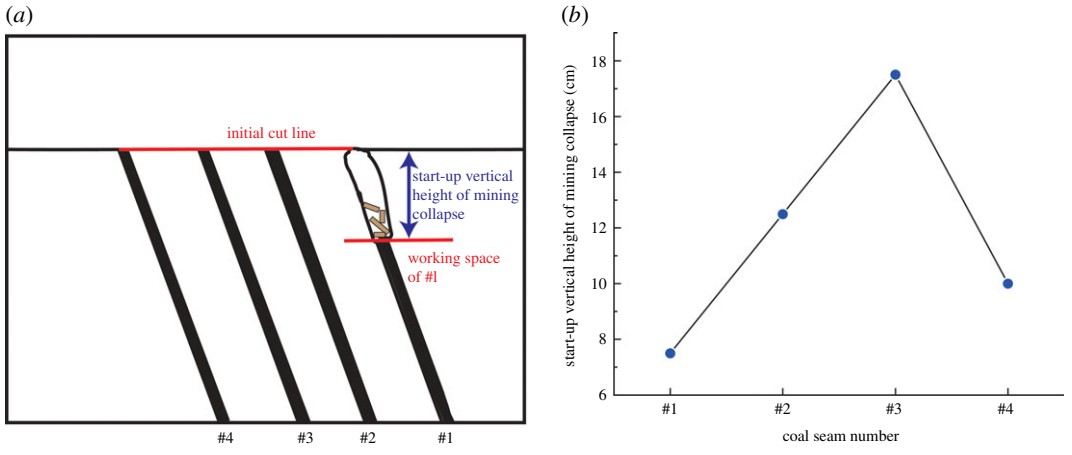

**Figure 3.** Start-up vertical height of immediate roof collapse in each coal seam.

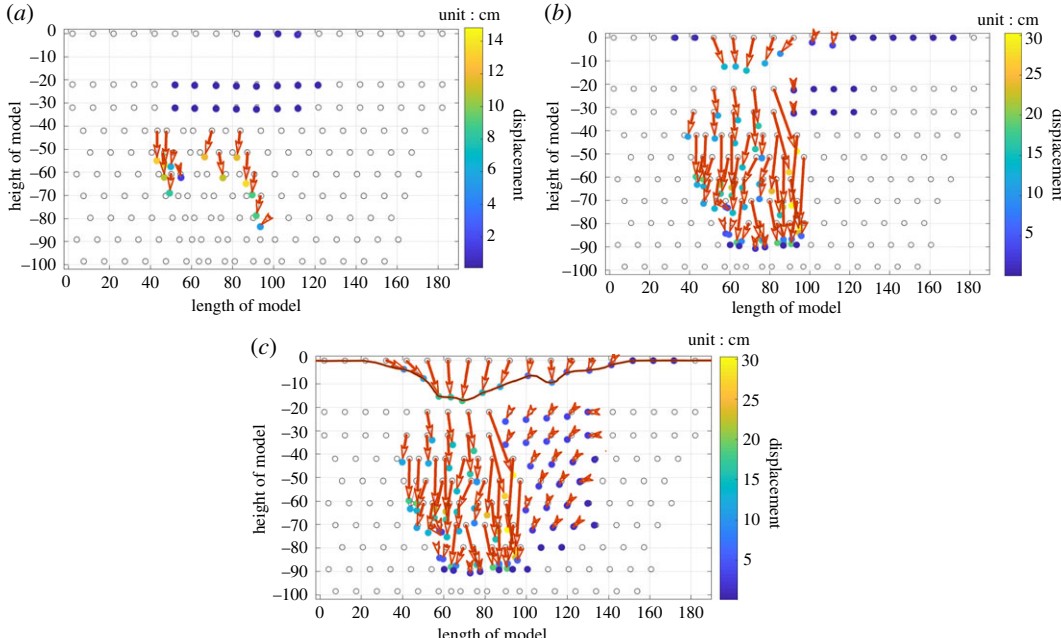

**Figure 4.** Displacement vector diagram of monitoring points at 5 h (*a*), 11 h (*b*) and 21 h (*c*) after mining completion.

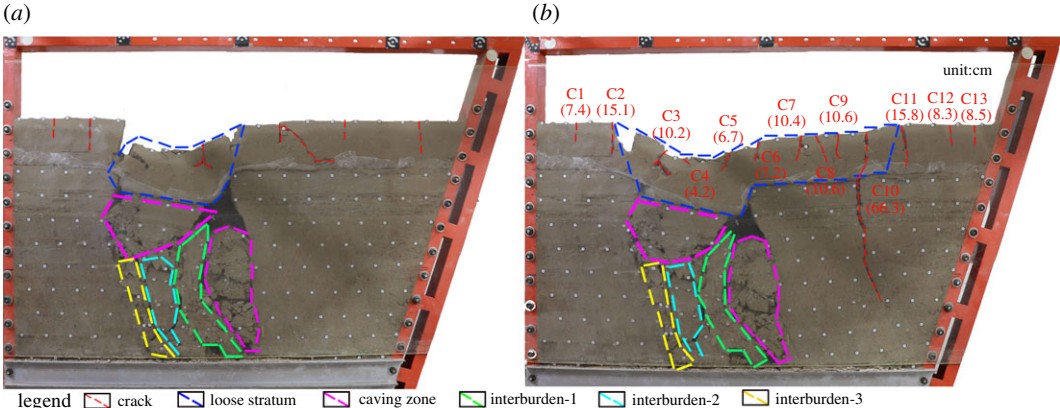

legend ⬜ crack ⬜ loose stratum ⬜ caving zone ⬜ interburden-1 ⬜ interburden-2 ⬜ interburden-3

**Figure 5.** Characteristics of overburden failure due to sudden collapse of the model 11 h (*a*) and 21 h (*b*) after the completion of mining ('C' denotes 'Crack', and the number in parentheses represents the length of the cracks).

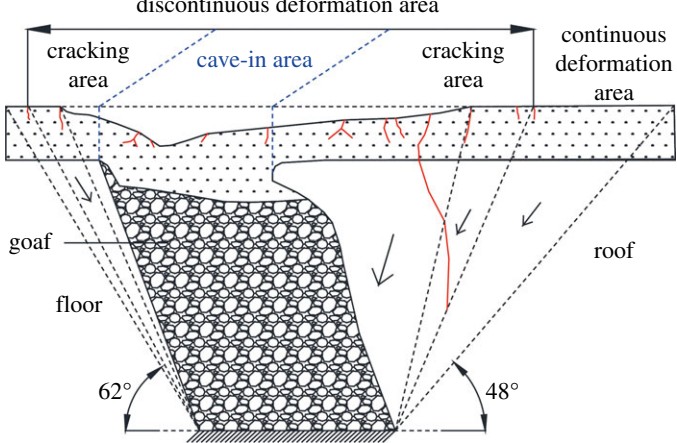

**Figure 6.** Schematic diagram of the overburden failure zoning after mining completion.

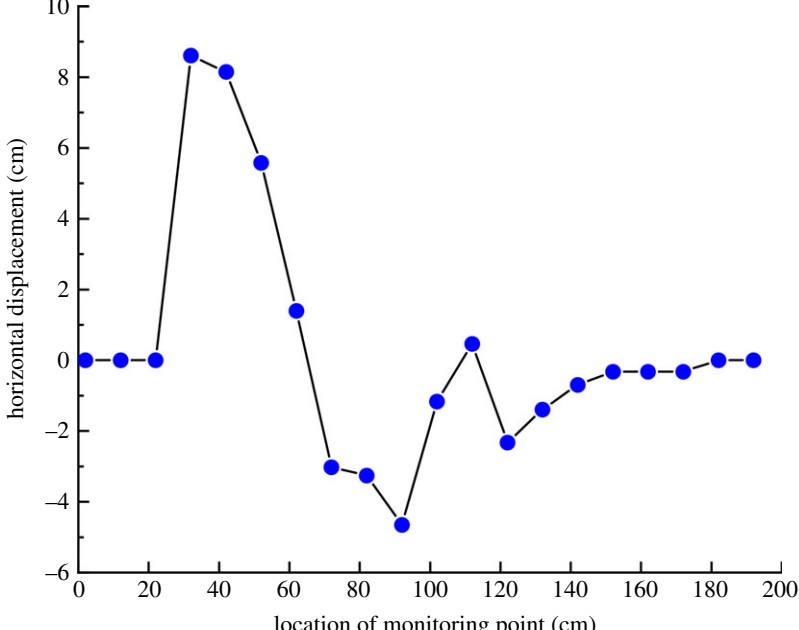

**Figure 7.** Horizontal displacement of surface monitoring points.

an extension of approximately 66.3 cm. At this point, a total of 13 new cracks appeared on the ground surface. Sixty hours after the completion of mining, the model gradually stabilized.

### 3.2.2. Surface cracks and horizontal deformation after mining completion

As shown in figure 6, the surface deformation zone can be divided into two regions: discontinuous deformation area and continuous deformation area. The discontinuous deformation area was characterized by large surface disturbances, such as tension cracks, steps and collapse pits. The tension cracks were formed by tensile deformation and developed along the nearly vertical discontinuities parallel to the coal seam (the main joint group). Under the action of stress relaxation, the ground with large vertical deformation formed a collapse; therefore, the discontinuous deformation area can be further divided into cave-in area and cracking area. Outside the discontinuous deformation area was the continuous deformation area, and the surface was less affected by mining.

It can be seen from figure 7 that after the model was stabilized, the maximum positive horizontal displacement occurred at 40 cm, which was located above coal seam #3. The maximum reverse horizontal displacement occurred at 90 cm, which was located at coal seam #1. Moreover, the two maximum displacement points were located at the edge of the cave-in area.

After the model was stabilized, the surface cracks were numbered C1–C13 from left to right (figure 5*a*). It can be seen from figure 5 that all the cracks were vertical tensile cracks, and the number of tensile cracks in the downward direction (C5–C13) was far greater than that in the upward direction (C1–C3). From C1 to C13, the length of the cracks first increased, then decreased (because C4 basically developed in the centre of the goaf), then increased and then decreased again. This pattern is similar to the length development patterns of surface cracks after horizontal coal seam mining. The displacement angle of the steeply inclined coal seam roof was measured at 48° and that in the floor was 62°. It can be seen that the range of surface influence along the roof direction of the coal seam is larger than that along the floor direction. In actual mining, attention should be paid to the control of surface subsidence along the roof direction.

# 4. Discussion

Previous studies have shown that in steeply inclined single-seam mining, the roof will gradually collapse, causing the surrounding rocks in the downhill direction to violently squeeze into the goaf, and the surface subsidence basin is biased in the descending slope direction. In this experiment, in addition to the phenomena induced by single-seam mining, the structure of the surrounding rock has changed dramatically due to coal seam mining in the previous stage, the more complex movement and failure processes of the rock strata in the later stage, and the more serious damage to the rock strata and surface.

According to the experimental phenomena, we believe that the dynamic change in the stope structure is the key to strata movement in steeply inclined multi-seam mining. Therefore, the change in the stope structure of steeply inclined multi-seam mining can be divided into two stages: a stable rock mass stage and a severe failure stage.

## 4.1. Stable rock mass stage

As shown in figure 8, when the first coal seam is excavated, the rock mass at the initial cut loses its support and collapses under the action of gravity. With the advance of the working face, the overhanging length of the roof also becomes longer, and the 'bending–cracking–falling–stacking' process gradually occurs. Due to the large inclination angle of the goaf, the broken rock mass generally accumulates near the working face, which has a great adverse effect on mining. It can be seen from the experimental phenomena that the deformation of the surrounding rock mainly includes the downward movement of the rock mass above the initial cut and the lateral movement of the roof strata. The movement of the rock mass in these two directions forms the bending area and supporting area, as shown in figure 8, thus forming a stable arch structure over the goaf to prevent the surrounding rock from further destruction.

Due to the arch structure mentioned above, the supporting area will also bear the weight of the rock mass within the range of the rock movement angle $\varphi$, so a decompression area will be formed at the roof of the second coal seam. This outcome also leads to roof cracking and small-scale collapse under self-weight during mining the second coal seam, which is generally stable, and the start-up vertical height of the mining collapse of the second coal seam is larger than that of the first coal seam. The same process occurs during the mining of the third and fourth coal seams. After a certain distance of group coal mining, the interburdens become supporting pillars, and the goaf presents a stable room-and-pillar support structure. Due to the inclined form of the interburdens, the overburden load applied at the initial cut makes the interburdens slanted compression pillars.

## 4.2. Severe failure stage

When the multi-seam mining reaches a certain extent, the arch structure of the first coal seam is damaged, and the original arch foot (figure 8) is unloaded, which causes the immediate roof of the first coal seam to slide toward the goaf. In the early stage of slippage, the moving speed is slow, and the change in the goaf is also slow (at this point, the stope structure is still in the stability stage). When the displacement caused by the slippage reaches a certain degree, the distribution of the overburden load will become increasingly uneven, which will accelerate the rock mass sliding. At this point, the stope structure enters the violent destruction stage of the rock mass, the goaf collapses extensively, and the rock mass structure is unstable.

As the roof of each seam has a larger overhanging area and a larger inclination angle, the collapsed immediate roof of the first coal seam (Area-A) will directly crush interburden-1, followed by interburden-2 and interburden-3, resulting in a 'domino' destruction effect. The interburdens in the goaf were

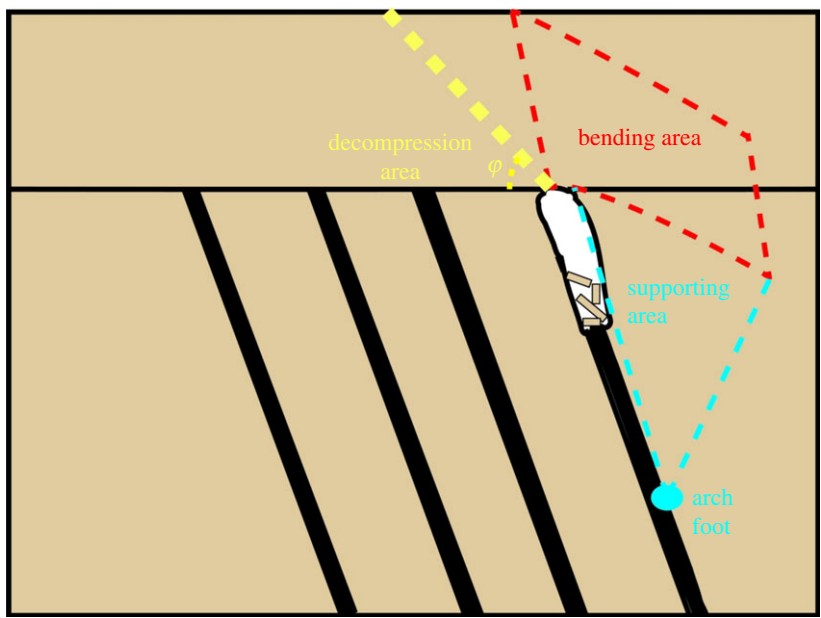

**Figure 8.** Schematic diagram of the arch structure of surrounding rock after mining the first layer of coal for a certain length (the range in the diagram is only indicative, not representative of the real scale).

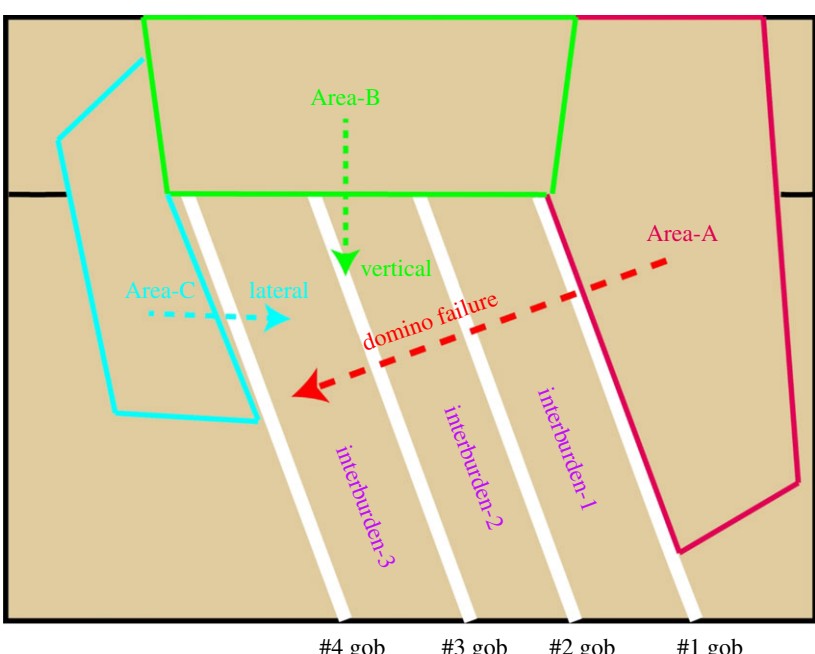

**Figure 9.** Schematic diagram of domino effect destruction in the goaf (the range in the figure is only a schematic and does not represent the real scale).

originally used to support the weight of the overburden strata. After the destruction of these interburdens, the overburden collapses, which leads to violent destruction to the surface.

From the failure zoning diagram (figure 9), it can be observed that Area-A is mainly the area squeezed and slipped into the goaf, Area-B is the area displaced downward and Area-C is the lateral squeeze area. Therefore, Area-B has the largest degree of damage and the most severe discontinuous deformation, and Area-A is second, but the scope of the damage is the most extensive. Area-C has a smaller degree of damage than the two areas above. This kind of damage is much more complicated than that caused by steeply inclined single-seam mining.

# 5. Conclusion

Through a physical model experiment, this paper simulates the process of steeply inclined multi-seam mining, analyses the dynamic change in goaf-rock mass structure in the mining process, summarizes the deformation and failure characteristics of overburden and divides the overburden into several areas. The main conclusions are as follows.

(1) In the initial period of steeply inclined coal seam mining, only a small amount of roof strata collapse occurs, and the start-up vertical height of mining collapse of the coal seam that was later mined is larger than that of the coal seam mined first. The collapsed rock mass is piled up near the working face, which has a great adverse effect on production.

(2) The stope structure can be divided into the stability stage and severe failure stage of the rock mass during the mining process. When the overhang area of the coal group roof does not reach a certain range, the interburdens bear the weight of the overburden rock mass, and the structure of the goaf is stable. When the overhang area of the coal group roof reaches a certain degree, the displacement of the rock mass in the descending direction will accelerate the stope structure into the severe failure stage, resulting in sliding failure and crushing the interburden in the form of a domino effect displacement, which causes the overburden to collapse and results in serious damage to the ground surface.

(3) After the completion of mining, the surface subsidence basin exhibits an asymmetric shape, and the range of destruction in the downhill direction is much larger than that in the uphill direction. According to the degree of ground surface damage, the surface discontinuous deformation area can be divided into a cave-in area and cracking area.

Data accessibility. Qin, Yan (2021), experimental data, Dryad, Dataset: https://doi.org/10.5061/dryad.j6q573nd3 [35].
Authors' contributions. Conceptualization was done by H.W. and Y.Q.; data curation was done by H.W. and Y.C.; formal analysis was done by H.W. and H.W.; investigation was done by X.L.; methodology was done by H.W. and H.W.; writing original draft was done by Y.C.; writing review and editing was done by H.W. and Y.Q.
Competing interests. We declare we have no competing interests.
Funding. This research is supported by the National Natural Science Foundation of China (grant no. 41772326).

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
