## [Peer Review File · Royal Society Open Science]

Review History

RSOS-210275.R0 (Original submission)

Review form: Reviewer 1

Is the manuscript scientifically sound in its present form?

Yes

Are the interpretations and conclusions justified by the results?

Yes

Is the language acceptable?

Yes

Do you have any ethical concerns with this paper?

No

Have you any concerns about statistical analyses in this paper?

No

Recommendation?

Accept with minor revision (please list in comments)

Comments to the Author(s)

I really enjoyed this paper. It is unusual nowadays to see a physical model study. However, they can provide a very clear depiction of the failure mechanics, as is the case in this study. My comments come under two heads. On the technical side, I think the paper would benefit from a more detailed discussion of the similitude analysis that was employed in to scale the model material properties. Also, it seems to me that Table 1 must be presenting the material properties that the author's were trying for. I can't believe that they were truly able to match the friction angle to one-tenth of a degree for example. Table 1 should also present the actual strengths and other properties that were actually achieved. Also, what was the thickness of the model (the thickness in the lateral direction, along the strike of the coal seams)? And what kind of material (if any) was used to constrain the model in the lateral direction?

My other comments have to do with the English translation. The translation is smooth and readily understandable for the most part. But some technical terms are unclear. For example, I think that the "open off cut" could be better translated as the "initial cut." A "small scope collapse" is probably a "small scale collapse," and the "direct roof" is the "immediate roof." The "start up vertical height of direct roof collapse" (see figure 3) is a very confusing term. I think you mean the vertical distance between the initial cut and face when the first collapse of the immediate roof occurs, but I am not sure. You should probably illustrate what you mean with a sketch. Lastly, the word "dehiscence" is a medical term that most English speakers have never heard of (I had to look it up). I think a better term might be "fracture zone." That term has the advantage that it fits with the well-known zones of overburden deformation above full extraction mining: the caving zone, the fracture zone, and the continuous deformation zone (see Prof. Syd Peng's work for example.)

Review form: Reviewer 2

Is the manuscript scientifically sound in its present form?

Yes

Are the interpretations and conclusions justified by the results?

Yes

Is the language acceptable?

Yes

Do you have any ethical concerns with this paper?

No

Have you any concerns about statistical analyses in this paper?

No

Recommendation?

Accept with minor revision (please list in comments)

Comments to the Author(s)

1. What's the ratio of the model (1:100 or something else), please explain?
2. Did the physical mode simulate real overburden at the mine? If you simulated the coal seams at "Balitang mine", please provide the actual overburden and the simulated overburden in the physical model.

3. Did the mining sequences in the physical model simulate the real mining sequence at the mine?
4. The coal seams thickness that you mentioned at the mine are 3.55m, 2.5m, 2.7m and 2.2m respectively, however, the coal seam thickness didn't look proportional in the physical model (# 1 and #2 apparently looks thicker than #3 and #4), please explain.
5. Please provide more explanation for the concept "Start-up vertical height of mining collapse". Since you have the measured numbers, please provide figures to show the measurement on the model for each seam. In addition, please explain how the "Start-up vertical height of mining collapse" will impact the real mining.
6. The paper mentioned "the displacement angle of the steeply inclined coal seam" in the physical model. Please explain what's the purpose to measure this angle and how this "angle" will impact the real mining.
7. This physical model is just one case with a specific geological setting (seam thickness, interburden, overburden and inclined angle etc.), what's the limitation of this model? Will different geological setting show different results, e.g. failure process and surface deformation, in the physical model and how?

Decision letter (RSOS-210275.R0)

Dear Dr Qin

On behalf of the Editors, we are pleased to inform you that your Manuscript RSOS-210275 "Process of overburden failure in steeply inclined multi-seam mining: insights from physical modeling" has been accepted for publication in Royal Society Open Science subject to minor revision in accordance with the referees' reports. Please find the referees' comments along with any feedback from the Editors below my signature.

Please submit your revised manuscript and required files (see below) no later than 7 days from today's (ie 19-Apr-2021) date. Note: the ScholarOne system will 'lock' if submission of the revision is attempted 7 or more days after the deadline. If you do not think you will be able to meet this deadline please contact the editorial office immediately.

on behalf of Professor Zach Agioutantis (Associate Editor) and R. Kerry Rowe (Subject Editor)
 openscience@royalsociety.org

Associate Editor Comments to Author (Professor Zach Agioutantis):

Associate Editor: 1

Comments to the Author:

Also please note that the units in the y-axis caption of Fig 3 should follow the international standard: it should be (cm) and NOT /cm

Reviewer comments to Author:

Reviewer: 1

Comments to the Author(s)

I really enjoyed this paper. It is unusual nowadays to see a physical model study. However, they can provide a very clear depiction of the failure mechanics, as is the case in this study. My comments come under two heads. On the technical side, I think the paper would benefit from a more detailed discussion of the similitude analysis that was employed in to scale the model material properties. Also, it seems to me that Table 1 must be presenting the material properties that the author's were trying for. I can't believe that they were truly able to match the friction angle to one-tenth of a degree for example. Table 1 should also present the actual strengths and other properties that were actually achieved. Also, what was the thickness of the model (the thickness in the lateral direction, along the strike of the coal seams)? And what kind of material (if any) was used to constrain the model in the lateral direction?

My other comments have to do with the English translation. The translation is smooth and readily understandable for the most part. But some technical terms are unclear. For example, I think that the "open off cut" could be better translated as the "initial cut." A "small scope collapse" is probably a "small scale collapse," and the "direct roof" is the "immediate roof." The "start up vertical height of direct roof collapse" (see figure 3) is a very confusing term. I think you mean the vertical distance between the initial cut and face when the first collapse of the immediate roof occurs, but I am not sure. You should probably illustrate what you mean with a sketch. Lastly, the word "dehiscence" is a medical term that most English speakers have never heard of (I had to look it up). I think a better term might be "fracture zone." That term has the advantage that it fits with the well-known zones of overburden deformation above full extraction mining: the caving zone, the fracture zone, and the continuous deformation zone (see Prof. Syd Peng's work for example.)

Reviewer: 2

Comments to the Author(s)

1. What's the ratio of the model (1:100 or something else), please explain?
2. Did the physical model simulate real overburden at the mine? If you simulated the coal seams at "Balitang mine", please provide the actual overburden and the simulated overburden in the physical model.
3. Did the mining sequences in the physical model simulate the real mining sequence at the mine?

4. The coal seams thickness that you mentioned at the mine are 3.55m, 2.5m, 2.7m and 2.2m respectively, however, the coal seam thickness didn't look proportional in the physical model (# 1 and #2 apparently looks thicker than #3 and #4), please explain.
5. Please provide more explanation for the concept "Start-up vertical height of mining collapse". Since you have the measured numbers, please provide figures to show the measurement on the model for each seam. In addition, please explain how the "Start-up vertical height of mining collapse" will impact the real mining.
6. The paper mentioned "the displacement angle of the steeply inclined coal seam" in the physical model. Please explain what's the purpose to measure this angle and how this "angle" will impact the real mining.
7. This physical model is just one case with a specific geological setting (seam thickness, interburden, overburden and inclined angle etc.), what's the limitation of this model? Will different geological setting show different results, e.g. failure process and surface deformation, in the physical model and how?

===PREPARING YOUR MANUSCRIPT===

===PREPARING YOUR REVISION IN SCHOLARONE===

Author's Response to Decision Letter for (RSOS-210275.R0)

See Appendix A.

Decision letter (RSOS-210275.R1)

Dear Dr Qin,

I am pleased to inform you that your manuscript entitled "Process of Overburden Failure in Steeply Inclined Multi-seam Mining: Insights from Physical Modeling" is now accepted for publication in Royal Society Open Science.

on behalf of Professor Zach Agioutantis (Associate Editor) and R. Kerry Rowe (Subject Editor)
openscience@royalsociety.org

Follow Royal Society Publishing on Twitter: @RSocPublishing
Follow Royal Society Publishing on Facebook:
<https://www.facebook.com/RoyalSocietyPublishing.FanPage/>

Read Royal Society Publishing's blog:
<https://royalsociety.org/blog/blogsearchpage/?category=Publishing>

Appendix A

Responses to Editor's and Reviewers' Comments

Yan Qin

School of Engineering and Technology
China University of Geosciences (Beijing)
qinyancugb@cugb.edu.cn

Acknowledgement The authors are grateful to the anonymous reviewers for a careful checking of the details and for helpful comments that improved this paper.

Responses to Editor's Comments

Comment # 1:

Please note that the units in the y-axis caption of Fig 3 should follow the international standard: it should be (cm) and NOT /cm.

Response:

Thanks for your careful review. We have already corrected '/cm' as '(cm)' in Fig.3 and Fig.7.

Responses to Reviewer # 1

Comment # 1:

On the technical side, I think the paper would benefit from a more detailed discussion of the similitude analysis that was employed in to scale the model material properties. Also, it seems to me that Table 1 must be presenting the material properties that the author's were trying for. I can't believe that they were truly able to match the friction angle to one-tenth of a degree for example. Table 1 should also present the actual strengths and other properties that were actually achieved. Also, what was the thickness of the model (the thickness in the lateral direction, along the strike of the coal seams)? And what kind of material (if any) was used to constrain the model in the lateral direction?

Response:

Your suggestions are very correct.

Firstly, we added the content of calculating the similarity constant of this experiment as shown in Page 4 with red color.

Secondly, the parameters listed in Table 1 are our target values. Therefore, we explained how to obtain the expected mechanical parameters of the model material in Page 4 (the words marked in red and underlined).

Thirdly, the thickness in the lateral direction of the model is 10cm, and the front and back of the model are fixed with acrylic plates. This sentence was also added into revised manuscript in Page 5 with red words.

Comment # 2:

My other comments have to do with the English translation. The translation is smooth and readily understandable for the most part. But some technical terms are unclear. For example, I think that the "open off cut" could be better translated as the "initial cut." A "small scope collapse" is probably a "small scale collapse," and the "direct roof" is the "immediate roof."

Response:

Thanks for your careful review and suggestions.

- (1) The “open-off cut” has been replaced with “initial cut” in full-text.
- (2) The “small scope collapse” has been replaced with “small scale collapse” in Page 11.
- (3) The “direct roof” has been replaced with “immediate roof” in full-text.

Comment # 3:

The "start up vertical height of direct roof collapse" (see figure 3) is a very confusing term. I think you mean the vertical distance between the initial cut and face when the first collapse of the immediate roof occurs, but I am not sure. You should probably illustrate what you mean with a sketch.

Response:

Thanks for your careful review. We have illustrated what we mean with a sketch and added it into Fig.3 as follow:

Fig. 3. Start-up vertical height of immediate roof collapse in each coal seam

Comment # 4:

Lastly, the word "dehiscence" is a medical term that most English speakers have never heard of (I had to look it up). I think a better term might be "fracture zone." That term has the advantage that it fits with the

well-known zones of overburden deformation above full extraction mining: the caving zone, the fracture zone, and the continuous deformation zone (see Prof. Syd Peng's work for example.)

Response:

Your suggestion is very responsible, and the authors are very appreciate.

However, we think the caving zone and the fracture zone are exclusive words used to describe deformation of overburden when a horizontal or gently inclined coal seam is being mined. In steeply inclined coal seam mining, these two exclusive terms may not fit well for the deformation characteristics are different with mining a horizontal or gently inclined coal seam.

We have redefined the dehiscence area as cracking area to avoid this misunderstanding and correspond with 'C' in Fig.5, as shown in Fig.6 and Page 15.

Responses to Reviewer # 2

Comment # 1:

What's the ratio of the model (1:100 or something else), please explain?

Response:

The geometric similarity of the model is 250 which is stated in Page 4. Besides, in the revised manuscript, we have added detailed information of other similarity constants into Page 4.

Comment # 2:

Did the physical mode simulate real overburden at the mine? If you simulated the coal seams at "Balitang mine", please provide the actual overburden and the simulated overburden in the physical model.

Response:

Yes. According to the real geological model, we reduced it according to the similarity ratio and established the similarity model. Because this paper focuses on the process of rock failure in the process of model test, the real model is not introduced too much. Relevant lithology and parameter information are listed in Table 1 and figured in Fig.1.

Comment # 3:

Did the mining sequences in the physical model simulate the real mining sequence at the mine?

Response:

Yes. The mining sequences were introduced in Fig.1.

Comment # 4:

The coal seams thickness that you mentioned at the mine are 3.55m, 2.5m, 2.7m and 2.2m respectively, however, the coal seam thickness didn't look proportional in the physical model (# 1 and #2 apparently looks thicker than #3 and #4), please explain.

Response:

Thanks for your careful review. We have checked the figure of physical model carefully, and no mistake were found. Since the geometric similarity of the model is 250, the thickness difference between #1 and #4 coal seam is only about 5mm in physical model. We think this is maybe a visual error.

Comment # 5:

Please provide more explanation for the concept "Start-up vertical height of mining collapse". Since you have the measured numbers, please provide figures to show the measurement on the model for each seam. In addition, please explain how the "Start-up vertical height of mining collapse" will impact the real mining.

Response:

Thanks for your careful review. We have illustrated what we mean with a sketch and added it into Fig.3 as follow:

Fig. 3. Start-up vertical height of immediate roof collapse in each coal seam

And how the "Start-up vertical height of mining collapse" will impact the real mining, we explain as follows:

It can be seen that in steeply inclined multi seam mining, the more backward the mining is, the larger the initial caving step distance of coal seam roof is. But in practice, the mechanical properties of rock mass should also be paid more attentions to prevent obvious mine pressure disasters.

The sentence have been added into revised manuscript in Page 12 with red words.

Comment # 6:

The paper mentioned "the displacement angle of the steeply inclined coal seam" in the physical model. Please explain what's the purpose to measure this angle and how this "angle" will impact the real mining.

Response:

Thanks for your careful review. How this "angle" will impact the real mining. We explain as follows:

It can be seen that the range of surface influence along the roof direction of coal seam is larger than that along the floor direction. In actual mining, attention should be paid to the control of surface subsidence along the roof direction.

The sentence have been added into revised manuscript in Page 16 with red words.

Comment # 7:

This physical model is just one case with a specific geological setting (seam thickness, interburden, overburden and inclined angle etc.), what's the limitation of this model? Will different geological setting show different results, e.g. failure process and surface deformation, in the physical model and how?

Response:

At present, the research on steeply inclined multi-seam is still relatively few, and the mechanism of strata movement can also be seen in relevant cases. This paper is the first time to use model test to study the process of strata movement and failure in steeply inclined multi-seam mining. The mechanism of strata movement in steeply inclined multi-seam mining needs to be further studied and summarized by multiple groups of numerical simulation tests and model tests in future.

The authors are grateful to the anonymous reviewers for helpful comments!